# Geodetic Mass Balance of the South Shetland Islands Ice Caps, Antarctica, from Differencing TanDEM-X DEMs

**Kaian Shahateet** [1,*] , **Thorsten Seehaus** [2] , **Francisco Navarro** [1] , **Christian Sommer** [2] **and Matthias Braun** [2]

1   Escuela Técnica Superior de Ingenieros de Telecomunicación, Universidad Politécnica de Madrid, 28040 Madrid, Spain; francisco.navarro@upm.es
2   Institut für Geographie, Friedrich-Alexander-Universität Erlangen-Nürnberg, D-91058 Erlangen, Germany; thorsten.seehaus@fau.de (T.S.); chris.sommer@fau.de (C.S.); matthias.h.braun@fau.de (M.B.)
*   Correspondence: k.fernandes@upm.es

**Abstract:** Although the glaciers in the Antarctic periphery currently modestly contribute to sea level rise, their contribution is projected to increase substantially until the end of the 21st century. The South Shetland Islands (SSI), located to the north of the Antarctic Peninsula, are lacking a geodetic mass balance calculation for the entire archipelago. We estimated its geodetic mass balance over a 3–4-year period within 2013–2017. Our estimation is based on remotely sensed multispectral and interferometric SAR data covering 96% of the glacierized areas of the islands considered in our study and 73% of the total glacierized area of the SSI archipelago (Elephant, Clarence, and Smith Islands were excluded due to data limitations). Our results show a close to balance, slightly negative average specific mass balance for the whole area of $-0.106 \pm 0.007$ m w.e. a$^{-1}$, representing a mass change of $-238 \pm 12$ Mt a$^{-1}$. These results are consistent with a wider scale geodetic mass balance estimation and with glaciological mass balance measurements at SSI locations for the same study period. They are also consistent with the cooling trend observed in the region between 1998 and the mid-2010s.

**Keywords:** SAR; remote sensing; glacier; Antarctic Peninsula; Antarctic periphery; ice loss

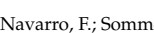



## 1. Introduction

One of the major impacts of climate change is sea level rise (SLR) due to the melting of land ice combined with thermal expansion of the oceans. The current contribution to SLR by the glaciers in the Antarctic periphery (Region 19—Antarctic and Subantarctic in the Glacier Regions classification [1]) is relatively small [2], but it is projected to increase substantially until the end of the 21st century [3]. The majority of its glacier area (63%) is situated in the Antarctic Peninsula (AP) region, where one of the most rapidly warming trends on Earth was observed during the second half of the 20th century [4–7]. However, Turner et al. [8] identified a turning point from a warming trend of 0.32 °C/decade during 1979–1997 to a cooling trend of $-0.47$ °C/decade during 1999–2014. They situated this turning point between mid-1998 and early 1999. Oliva et al. [9] showed that this cooling trend has been most significant in the N and NE of the AP and the South Shetland Islands (SSI) (Figure 1), with typical summer mean temperature changes of $-0.5$ °C between the decades 1996–2005 and 2006–2015. Oliva et al. [9] also analyzed the impact of this cooling trend on the cryosphere of the northern AP, including the slow-down of glacier recession, a shift to a slight positive surface mass balance of the peripheral glaciers, and, in the SSI, also a lengthening of the snow cover duration and a thinning of the active layer of permafrost. Carrasco et al. [10] have suggested that this cooling trend could have come to an end. In their analysis of the near-surface air temperature record for 1978–2020, they identified two breaking points. The first one matches with the turning point from warming to cooling in 1998/1999 described by [8]. The second has the opposite sign, indicating the return to a warming trend in the mid-2010s.

The South Shetland Islands are located in the northern sector of the AP region (Figure 1) where several glacier dynamics and mass balance studies relevant to our purposes have been undertaken. These include, among others, da Rosa et al. [11], who analyzed land-terminating glaciers in King George Island between 1956 and 2018 using remote sensing techniques. They estimated a shrinkage in the area of Tower, Windy, Ecology, Baranowski, and Sphinx Glaciers by 70, 31, 25, 25, and 21%, respectively. Osmanoğlu et al. [12] used the method of intensity offset tracking to evaluate the ice surface velocity and discharge in King George Island over 2008–2011. They estimated a total ice discharge of $720 \pm 428$ Mt a$^{-1}$. In another study, Osmanoglu et al. [13] determined the glacier surface velocities and their seasonal variations, as well as the ice discharge and the surface mass balance of the Livingston Island ice cap, for the period 2007–2011. They also estimated the total mass balance during that period by combining ice discharge to the ocean with the surface mass changes. The average surface mass balance of Livingston Island for this period was estimated as $0.06 \pm 0.14$ m w.e. a$^{-1}$, which, added to a frontal ablation of $-0.73 \pm 0.38$ m w.e. a$^{-1}$ (expressed in terms of specific mass loss over the ice cap area), gave a total specific mass balance of $-0.67 \pm 0.40$ m w.e. a$^{-1}$, equivalent to a mass change of $-467 \pm 279$ Mt a$^{-1}$ over the whole island. Pętlicki et al. [14] analyzed the surface elevation changes of Ecology Glacier, King George Island, for several periods within 1979–2016, using archival cartographic material and various in situ surveying and remote sensing techniques. Of particular relevance to our study are the surface elevation changes during 2012–2016, based on DEM differencing from a 2016 terrestrial laser scanning DEM and a 2012 DEM derived from a tri-stereo set of Pléiades 1A panchromatic images. For this period, they found a mean elevation change of $-0.5 \pm 0.6$ m a$^{-1}$. Another remote sensing study relevant to this paper is that of Fieber et al. [15], who performed an analysis of surface elevation and volume/mass changes of 16 individual glaciers, grouped at four locations across the Antarctic Peninsula, from stereo WorldView-2 satellite and archival aerial imagery. Of these 16 glaciers, 10 were located in the SSI (9 in King George Island and 1 in Elephant Island) and span the period 1956–2013, immediately preceding our study period of 2013–2017, thus providing the opportunity to analyze the evolution of the mass balance changes of such glaciers between both periods.

In addition to the above remote sensing-based calculations, several surface mass balance studies using the glaciological method have been undertaken in the SSI. These have been summarized by Navarro et al. [16] and include the measurements on the G1 Glacier in Deception Island, 1969–1974 [17]; Rotch Ice Dome in Livingston Island, 1971–1974 [17]; Nelson Island ice cap, 1986–1989 [18]; and King George Island ice cap, 1969–1971 [17] and 1985–1992 [19]. The study conducted by Navarro et al. [16] was focused on Hurd and Johnsons Glaciers, Livingston Island, over the period 2002–2011 (additional data for these two glaciers up until the present day are available on the World Glacier Monitoring Service database [20]).

As can be seen from the above, most of the mass balance studies in the SSI are at the local level; only one analysis covers a whole island ice cap (Livingston Island [13]), and another one covers all of the marine-terminating basins of an ice cap (King George Island [12]). On the opposite side, regional studies cover a much wider area and either have larger error bars (e.g., [21]) or do not address our zone of interest (the SSI) with sufficient detail (e.g., [2]). We aim to fill this gap by performing a calculation of the geodetic mass balance (GMB) of the SSI, for the period 2013–2017, with detail of its spatial distribution. Knowing the spatial distribution of the mass losses is essential to understand their underlying processes, for instance, whether the mass losses are dominated by dynamic thinning and iceberg calving or by surface mass balance. This study thus provides detailed data to both validate the results from wider scale (regional to global) studies of contribution of wastage from glaciers to sea level rise and to understand the various processes contributing to these mass losses under a changing climate.

We follow the methodology described in previous studies such as those of [22–24], which has shown its effectiveness in computing GMB from synthetic aperture radar (SAR)

images at a low cost. Although GMB has a wider temporal span and often a larger spatial extent than that provided by the glaciological method, both methods complement each other. For instance, mass balances calculated using the glaciological method can be used to validate GMB estimates, and GMB can be used to check if the glaciological method estimates are free of systematic errors [25].

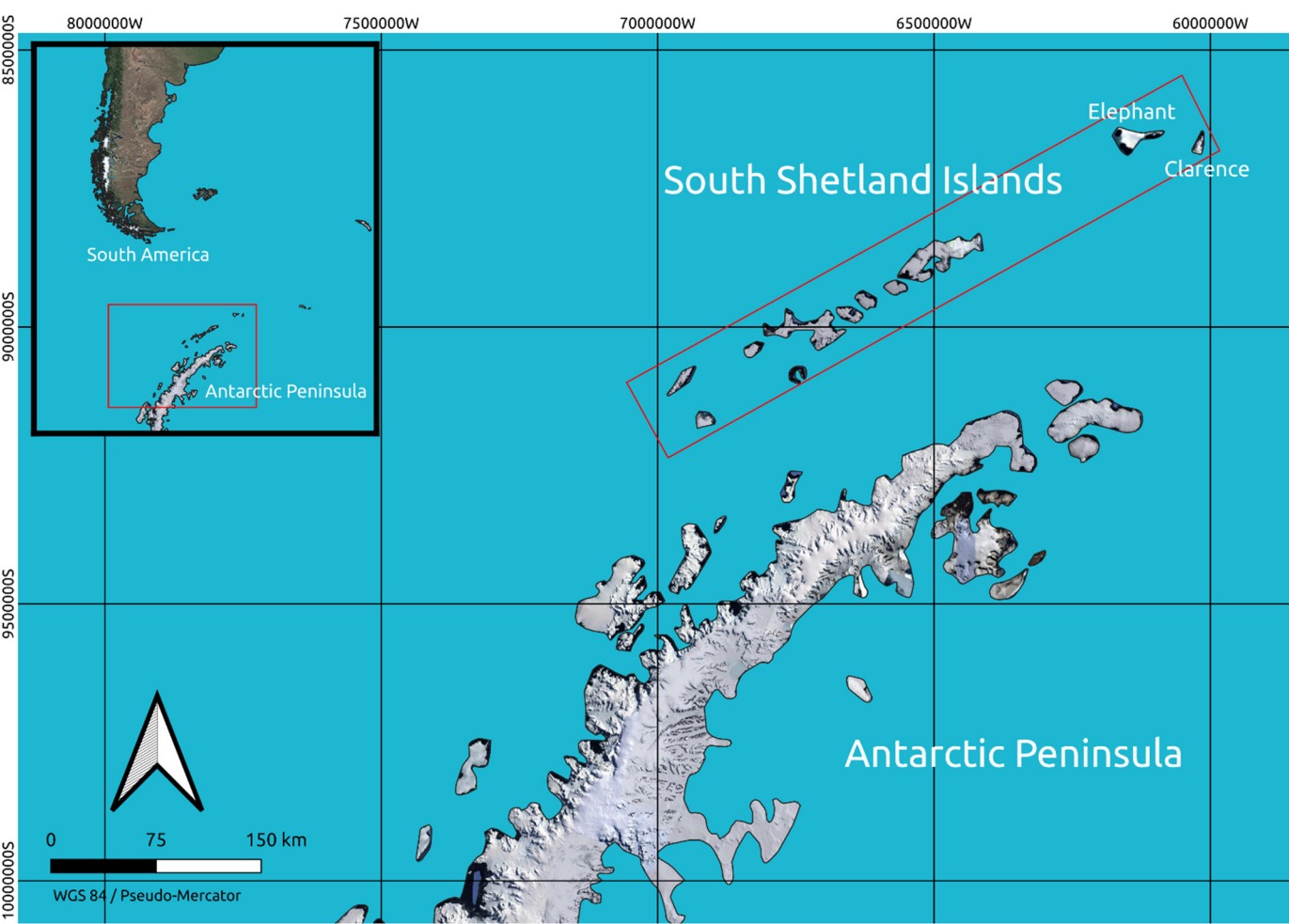

**Figure 1.** Overview of the study site. Background from Google Earth.

## 2. Materials and Methods

Our coverage of the SSI represents 73% of the total glacierized area of the archipelago. In regions with steep relief, the SAR images suffer shadows and layovers, which forced us to exclude Smith and Clarence Islands from our calculation. Although Elephant Island relief is not as steep, we could not calculate its GMB due to the lack of a good reference digital elevation model (DEM). Our study is therefore restricted to King George, Robert, Nelson, Greenwich, Livingston, Deception, Snow, and Low Islands, which, henceforth, we refer to as the South Shetland Islands, even if it excludes the three islands mentioned above. For this set of islands under study, our coverage is of 96% of the glacierized area.

The methodology to obtain geodetic mass balances used in the present work is based on [22–24]. It is structured into 4 main steps: DEM generation, DEM co-registration, height change calculation, and mass change computation.

**DEM generation:** We used data from the TerraSAR-X add-on for Digital Elevation Measurement mission (TanDEM-X) acquired in bi-static mode, operated jointly by the German Aerospace Center (DLR) and Astrium Defense and Space. To avoid variations in the glacier surface conditions that could lead to different radar penetrations, we selected data corresponding to the austral winter, from the beginning of May until the end of November,

when the liquid water content in the snowpack is expected to be minimum [23,24]. Two suitable coverages by TanDEM-X data were available for the SSI, one from 2014 (excluding King George Island, for which we used 2013 data) and another from 2017. Table A1 summarizes the data used in this study.

DEMs were generated by applying differential interferometric synthetic-aperture radar (DInSAR) techniques on the co-registered single look slant range complex (CoSSC) TanDEM-X data [23], which require the availability of a reference DEM. For Deception, Greenwich, Low, Nelson, Robert, and Snow Islands, we used the Reference Elevation Model of Antarctica (REMA) [26]. For Livingston Island, we used a DEM developed by Étienne Berthier (personal communication, 6 June 2020) from Pléiades stereoscopic imagery using NASA's Ames Stereo Pipeline. For King George Island, we used a DEM derived from TanDEM-X imagery by Sánchez-Gámez (personal communication, 27 May 2020). The reference DEMs were corrected to remove gaps and jumps; areas lacking values were filled by applying bi-linear interpolation using GIS tools, such as QGIS and R.

To obtain DEMs from the TanDEM-X data, we first concatenated scenes in the along-track direction from the same acquisition date and track. Next, differential interferograms were generated from the SAR data and the reference DEMs. This was followed by Goldstein filtering [27] of the interferograms and phase unwrapping procedures (using the minimum cost flow approach) and a conversion of the differential phase into differential height. Finally, the elevation information of the reference DEM was added to the differential heights to produce the final DEM.

**DEM co-registration:** In order to compute high-quality GMBs, horizontal or vertical shifts between consecutive DEMs should be minimal. For this reason, we first carried out a co-registration over stable ground of the generated DEMs from the earliest coverage (2013 and 2014) and the reference DEMs to generate a smooth mosaic of TanDEM-X DEMs. The stable ground was defined by selecting land areas without ice cover using Landsat images and glacier outlines from the Randolph Glacier Inventory [28] and from Silva et al. [29]. A further requirement for stable ground was an upper bound for a slope of 50°.

To reduce the shifts between both DEMs, we performed a vertical bi-linear co-registration on the stable areas of the TanDEM-X DEMs relative to the reference DEMs. The resulting DEMs were horizontally co-registered to the reference DEMs following the procedure of Nuth and Kääb [30]. The vertically and horizontally co-registered TanDEM-X DEMs were again vertically co-registered to remove the remaining vertical offsets. Finally, the resulting TanDEM-X DEMs were mosaicked into single DEMs for each island, including a timestamp layer of the individual pixels.

**Height change calculation:** From the previous step, we obtained TanDEM-X DEM mosaics for each island for 2013/14 and 2017. This was followed by another vertical and horizontal co-registration procedure, now using the TanDEM-X DEM mosaics from 2013/14 as the reference DEM (master image) and the DEM of 2017 as the one to be adjusted (slave image). By subtracting the resulting DEM mosaics, we generated an elevation change ($\Delta h$) map. From this $\Delta h$ map and the time difference of the individual pixels ($\Delta t$ map), we calculated the elevation change rate ($\Delta h/\Delta t$) maps for each island. Areas with slopes greater than 50° were masked out since the DEMs are less reliable on these areas and nearly no ice is accumulated there (avalanche slope).

Gaps in the $\Delta h/\Delta t$ maps were filled by using mean hypsometric interpolation at 100 m elevation bands [31]. The $\Delta h/\Delta t$ values in each bin were filtered by applying the 1–99% quantile filter. This approach was selected by testing different filters (namely quantile filter, 3 times normalized median absolute deviation, and no filter at all) as evaluated by Sommer et al. [24].

**Mass change computation:** To calculate the geodetic mass balance rates ($\Delta M/\Delta t$), we followed the recommendation of Cogley et al. [32]. Essentially, we integrated the elevation change rate all over the glacier areas (inventory adapted with Pléiades images from Pfeffer et al. and Silva et al. [28,29] of the individual islands to obtain volume change rates ($\Delta V/\Delta t$)

and multiplied the resulting volume change rate by a volume-to-mass conversion factor $f_{\Delta V}$ [33]):

$$\Delta M / \Delta t = f_{\Delta V} \Delta V / \Delta t \tag{1}$$

$f_{\Delta V}$ can span a wide range from 0 to 2000 kg m$^{-3}$ due to the possibility of different height changes in the accumulation and ablation zones, where the densities are substantially different. Here, we used the value of $850 \pm 60$ kg m$^{-3}$ recommended by [33] and also the commonly used value of $900 \pm 60$ kg m$^{-3}$. Because these values are very close to ice density, from now on, we refer to the conversion factor ($f_{\Delta V}$) as ice density ($\rho$).

**Uncertainty**

According to Seehaus et al. [23], the total uncertainty of the GMB results from the following:

- $\delta_{\Delta h / \Delta t}$: Error in surface elevation change rate resulting from DEM differencing. The uncertainty of hypsometric gap filling is included in this term.
- $\delta_S$: Error in glacier area (S) derived from the uncertainty in the glacier outlines.
- $\delta_\rho$: Uncertainty resulting from the volume-to-mass conversion using a fixed ice density.
- $V_{pen}/\Delta t$: Uncertainty due to the difference in radar signal penetration.

Therefore, applying error propagation, the total uncertainty is given by

$$\delta_{\Delta M / \Delta t} = \sqrt{\left( \left( \frac{\Delta M}{\Delta t} \right)^2 \left( \left( \frac{\delta_{\Delta h / \Delta t}}{\Delta h / \Delta t} \right)^2 + \left( \frac{\delta_S}{S} \right)^2 + \left( \frac{\delta_\rho}{\rho} \right)^2 \right) + \left( \frac{V_{pen}}{\Delta t} \rho \right)^2 \right)} \tag{2}$$

In general, $V_{pen}$ would be calculated by multiplying the difference in penetration depth between both measurements (arising from different environmental conditions—particularly surface melt—or the use of sensors with different frequencies) and the glacierized area. In our case, we used data from the same SAR sensor (TanDEM-X) corresponding to the same season (wintertime) in both campaigns. Consequently, we assumed zero difference in the penetration depth of the radar signal; therefore, $V_{pen}$ was set to zero. The same assumption was taken, e.g., by [23,24], on the AP. We also performed a backscattering analysis to evaluate the change in the reflectivity pattern between the two SAR data periods, individualized for each island, which could serve as an indicator for the different surface conditions and, thus, SAR signal penetration. However, the resulting patterns were sufficiently similar for both periods, verifying the approach that we employed.

For the uncertainty $\delta_\rho$, we used 60 kg m$^{-3}$ as suggested by [33]. For the area uncertainty, we used the $\pm 3\%$ value reported by [34], further applying a scaling factor accounting for different perimeter–area ratios as in [22]:

$$\frac{\delta_S}{S} = \frac{r_{p/s}}{r_{p/s \ Paul etal.}} 0.03 \tag{3}$$

where $r_{p/s}$ is the perimeter–area ratio per island, and $r_{p/s \ Paul etal.}$ is the perimeter–area ratio of Paul et al. [34].

We calculated the uncertainty due to DEM differencing ($\delta_{\Delta h / \Delta t}$) by analyzing the height change rate on stable areas (ice-free areas with a slope less than 50°) and used a 2–98% quantile filter to exclude outliers that would disproportionately contribute to the uncertainty calculation. The result was then aggregated in 5° slope bins to account for the dependence between surface slope and $\delta_{\Delta h / \Delta t}$, and the standard deviation for each bin ($\sigma_{\Delta h / \Delta t}$) was computed. Each slope bin was again filtered (2–98% quantile) to remove remaining artifacts. The $\delta_{\Delta h / \Delta t}$ of the elevation changes on glacierized areas was then estimated by weighting the obtained standard deviations for each slope bin on stable ground by the slope distribution on glacierized areas ($\sigma_{\Delta h / \Delta t \ AW}$). To include the uncertainty

due to auto-correlation in the error budget, we followed the approach from Rolstad et al. [25]. Accordingly, we estimated the accuracy of the elevation change as follows:

$$S_C = d_C^2 \pi$$

$$\delta_{\frac{\Delta h}{\Delta t}} = \sqrt{\frac{S_C}{5 S_G}} \sigma_{\frac{\Delta h}{\Delta t} AW} \text{ for } S_G > S_C \tag{4}$$

$$\delta_{\Delta h / \Delta t} = \sigma_{\Delta h / \Delta t \ AW} \text{ for } S_G > S_C$$

where $S_C$ is the spatial correlation area, and $S_G$ is the glacier area. The factor of 5 in Equation (4) was determined empirically by Rolstad et al. [25]. $d_C$ is the decorrelation distance (also known as lag distance) of the semivariogram. To generate the semivariograms, we calculated $\Delta h / \Delta t$ for 100,000 random samples on ice-free areas (stable ground) for each island and obtained an area-weighted mean decorrelation distance ($d_C$) of ~330 m. Equation (4) was applied for each continuous glacier area.

To account for the error due to hypsometric extrapolation, we applied the approach of Brun et al. [35], who multiply $\delta_{\Delta h / \Delta t}$ by a constant factor on the void-filled areas (we used a factor of 2 according to [22]).

## 3. Results

Over the period 2013–2017, the total area studied presented a mass change rate of $-238 \pm 12$ Mt a$^{-1}$ and a specific mass balance rate of $-0.106 \pm 0.007$ m w.e. a$^{-1}$, considering $\rho_{ice} = 850$ kg m$^{-3}$ or $-251 \pm 13$ Mt a$^{-1}$ and $-0.113 \pm 0.006$ m w.e. a$^{-1}$, respectively, if $\rho_{ice} = 900$ kg m$^{-3}$ was considered (Table 1).

Figure 2 shows both the mass change rate per island and its spatial distribution. We can see that Livingston and Deception are the only islands showing positive mass changes, with an order of magnitude larger for Livingston due to its much greater area, although Deception has a specific rate twice as large as that of Livingston (Table 1). By contrast, Low and Snow Islands have the most negative mass changes and also the most negative specific rates. Because of its size, King George Island, the largest island in the archipelago, also has a rather negative mass change in spite of its relatively small specific rate. Nelson Island, with a much smaller area than that of King George Island, shows a similar mass change rate due to its substantially larger specific rate. Finally, Robert and Greenwich present the least negative mass change rates, with specific rates also small, similar to that of King George Island.

Regarding the spatial distribution of the surface elevation change rates, Livingston presents important thinning rates and front retreat in Walker Bay and the Struma, Huron, and Huntress Glaciers (Figure 2). The largest thinning rates in King George Island are mostly focused on King George Bay (Figure 2). The elevation change rates are more heterogeneously distributed across Greenwich Island and mostly concentrated in the southeastern cost of Snow Island. Nelson, Robert, Low, and Deception show a relatively homogeneous distribution of elevation changes.

In addition to the elevation, volume, and mass changes per island, Table 1 provides information about the percentage of glacierized area covered by our GMB calculation. The overall coverage for the study islands is above 96%. Eight out of nine islands had at least 96% coverage. Only Livingston had a lower coverage, at ~90%. Considering the complete South Shetland archipelago (i.e., including Smith, Clarence, and Elephant Islands), the coverage is 73% of the glacierized area.

Figure 3 shows the hypsometric distribution of the glacierized areas and the corresponding distribution of elevation change rates ($\Delta h / \Delta t$) for each island and for the whole area of study (1–99% quantiles of $\Delta h / \Delta t$ of each elevation bin). Some elevation bins have strong deviations from the mean $\Delta h / \Delta t$ value of the corresponding island. Notably, Livingston has elevation bins above 1500 m a.s.l. with high departures from the mean and also with high uncertainties. We note that these high altitude bins have steep relief, which increases the uncertainty. However, as illustrated in Figure 3, these values lie within the

lower and upper 2% quantiles of the total glacierized measured area, not being relevant contributors to the mean calculation. The upper 2% shade also indicates that the bins above 700 m a.s.l. for the whole SSI (Figure 3i) do not strongly contribute to the mean $\Delta h/\Delta t$ value and its associated uncertainty calculation.

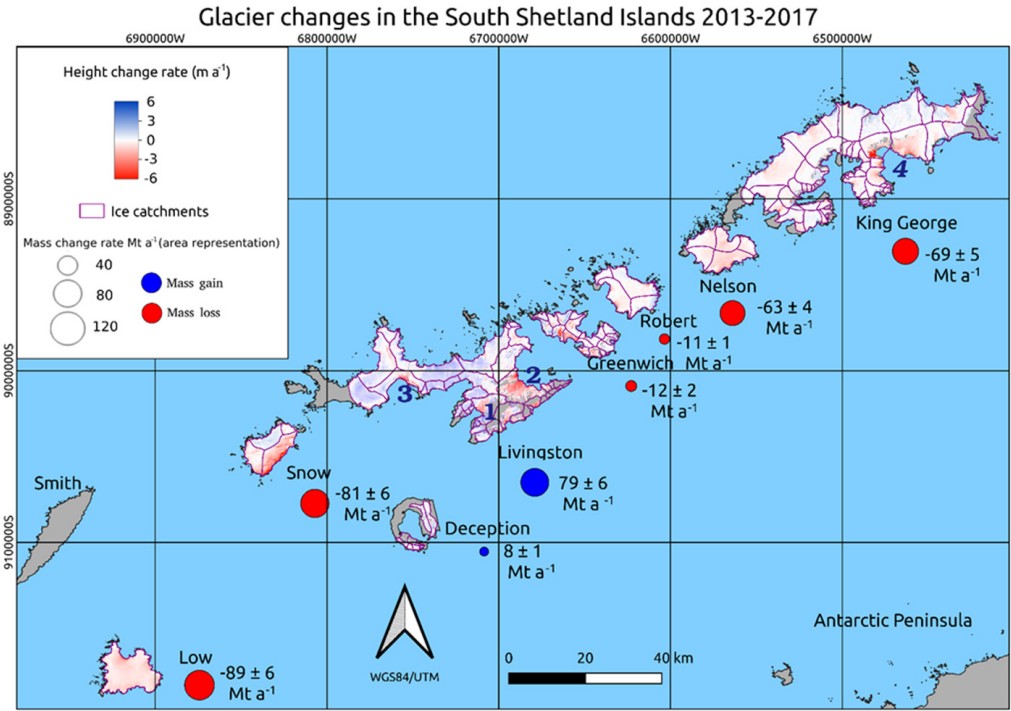

**Figure 2.** Spatial distribution of the surface elevation change rates (m a$^{-1}$) in the South Shetland Islands. The blue and red circles have areas proportional to the amount of mass gained or lost per year (Mt a$^{-1}$) by each island, respectively, calculated using $\rho_{ice}$ = 850 kg m$^{-3}$. Dark blue numbers indicate regions of interest cited in the text: 1—Huntress Glacier; 2—Huron and Struma Glaciers; 3—Walker Bay; and 4—King George Bay.

**Table 1.** Elevation change, volume change, mass change, and specific mass balance rates for the individual islands and for the whole set of islands studied, with indication of the observation period. Mass change and specific mass balance rates are given for two different density values.

| Island | Observation Period | Total Glacier Area (km$^2$) | Glacierized Area Covered by the GMB Calculation (%) | Mean Elevation Change (m a$^{-1}$) | Volume Change Rate (km$^3$ a$^{-1}$) | Mass Change Rate Considering $\rho_{ice}$ = 850 kg/m$^3$ (Mt a$^{-1}$) | Mass Change Rate Considering $\rho_{ice}$ = 900 kg/m$^3$ (Mt a$^{-1}$) | Specific Mass Balance Rate Considering $\rho_{ice}$ = 850 kg/m$^3$ (m w.e. a$^{-1}$) | Specific Mass Balance Rate Considering $\rho_{ice}$ = 900 kg/m$^3$ (m w.e. a$^{-1}$) |
|---|---|---|---|---|---|---|---|---|---|
| Livingston | 2014–2017 | 647,296 | 89.8 | 0.143 ± 0.004 | 0.092 ± 0.003 | 79 ± 6 | 83 ± 6 | 0.121 ± 0.009 | 0.128 ± 0.009 |
| Deception | 2014–2017 | 33,852 | 97.2 | 0.291 ± 0.021 | 0.010 ± 0.001 | 8 ± 1 | 9 ± 1 | 0.247 ± 0.026 | 0.262 ± 0.027 |
| Robert | 2014–2017 | 127,429 | 99.7 | −0.102 ± 0.006 | −0.013 ± 0.001 | −11 ± 1 | −12 ± 1 | −0.087 ± 0.008 | −0.092 ± 0.008 |
| Greenwich | 2014–2017 | 122,703 | 96.5 | −0.111 ± 0.012 | −0.014 ± 0.001 | −12 ± 2 | −12 ± 2 | −0.094 ± 0.012 | −0.100 ± 0.013 |
| Nelson | 2014–2017 | 142,214 | 99.9 | −0.525 ± 0.003 | −0.075 ± 0.000 | −63 ± 4 | −67 ± 5 | −0.446 ± 0.032 | −0.472 ± 0.032 |
| King George | 2013–2017 | 936,36 | 98.9 | −0.087 ± 0.001 | −0.081 ± 0.001 | −69 ± 5 | −73 ± 5 | −0.074 ± 0.005 | −0.078 ± 0.005 |
| Snow | 2014–2017 | 109,093 | 99.2 | −0.869 ± 0.006 | −0.095 ± 0.001 | −81 ± 6 | −85 ± 6 | −0.739 ± 0.052 | −0.782 ± 0.052 |
| Low | 2014–2017 | 125,135 | 99.7 | −0.836 ± 0.003 | −0.105 ± 0.001 | −89 ± 6 | −94 ± 6 | −0.710 ± 0.050 | −0.752 ± 0.050 |
| **Whole South Shetland Islands** | **2013–2017** | **2244,082** | **96.3** | **−0.125 ± 0.004** | **−0.281 ± 0.004** | **−238 ± 12** | **−251 ± 13** | **−0.106 ± 0.007** | **−0.113 ± 0.006** |

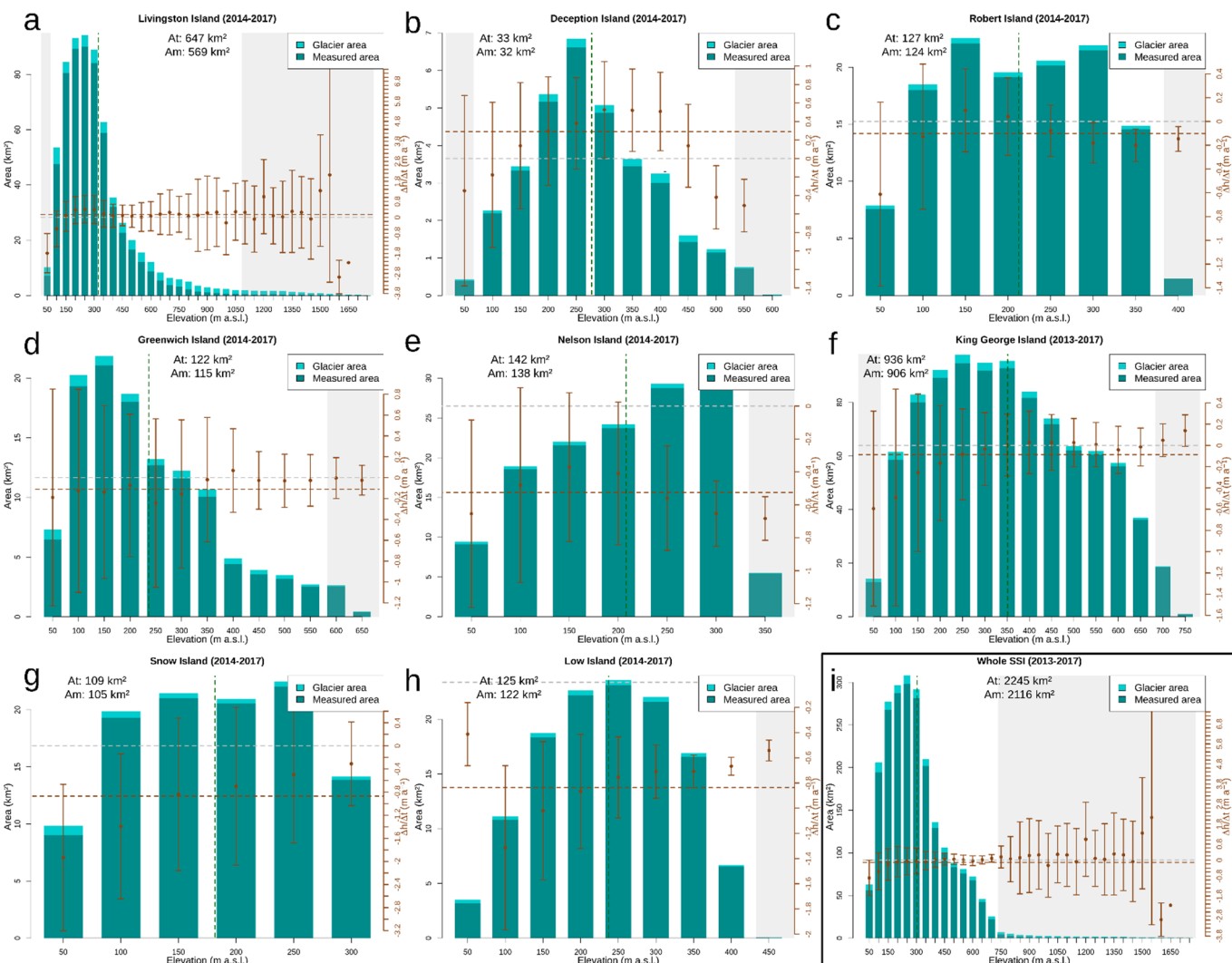

**Figure 3.** Hypsometric distribution of measured (dark blue bars) and total (light blue bars) glacier area of the individual islands and for the South Shetland archipelago. Note that the vertical axes scales are different. Brown dots represent the mean elevation change ($\Delta h / \Delta t$) in each elevation interval. Brown error bars represent the NMAD of the $\Delta h / \Delta t$ of each hypsometric bin. Grey areas indicate the lower and upper 2% quantiles of the total glacier area distribution. The vertical dark green dashed line is the mean elevation of each island. Areas with slope higher than 50° were masked out. The brown dashed line represents the mean elevation change rate of each island and the gray dashed line is the zero elevation change. At and Am indicate, respectively, the total and measured areas for each island.

## 4. Discussion

Our study covers 96.3% of the studied islands, similar to the 92.4% coverage of Hugonnet et al. [2] for the Antarctic and Subantarctic regions. If Elephant, Smith, and Clarence Islands are included, the coverage reduces to 73% of the total glacierized area of the South Shetland Islands, which is still a sufficient amount to consider our mass balance estimate as representative of the whole archipelago.

Our results show a mean elevation change of $-0.125 \pm 0.004$ m a$^{-1}$ or a specific mass balance of $-0.106 \pm 0.007$ m w.e. a$^{-1}$, representing a mass change of $-0.238$ Gt a$^{-1}$, for the period 2013–2017. Our specific mass change result for the SSI is in agreement with that of $-0.094 \pm 0.040$ m w.e. a$^{-1}$ found by Hugonnet et al. [2] for the Antarctic and Subantarctic regions (number 19) over the period 2015–2019, for which the total mass change was $-11.6$ Gt a$^{-1}$. The results also agree with those of Zemp et al. [21] for 2006–2016, of $-0.11 \pm 0.87$ m w.e. a$^{-1}$, and a corresponding mass change of $-0.14 \pm 108$ Gt a$^{-1}$, though we note that their error bars are huge, and their calculation is based on extrapolation of

observations covering <1% of the glacierized area. These values for Region 19 are small when compared to those of other regions, such as Alaska, which is currently the main contributor to sea level rise (excluding the ice sheets of Antarctica and Greenland), at $-87.2$ Gt a$^{-1}$ and a specific balance of $-1.054$ m w.e. a$^{-1}$. The second and third most important contributors are the Canadian Arctic South and North regions, with $-29.9$ and $-28.9$ Gt a$^{-1}$ ($-0.740$ and $-0.281$ m w.e. a$^{-1}$), respectively.

Although the current glacier mass losses of the Antarctic and Subantarctic regions (including the SSI) are modest, they are projected to increase substantially throughout the century. For instance, Edwards et al. [3], considering an intermediate Shared Social Pathway (SSP2) and an intermediate Representative Concentration Pathway (RCP 4.5), projected a 1.5 cm SLR contribution by the Antarctic and Subantarctic regions by 2100. This is close to the level projected for Alaska and the Canadian Arctic North, of 1.6 cm each. The Canadian Arctic South is expected to contribute 0.7 cm.

Our mass balance results close to balance are in agreement with the hiatus of warming in the Antarctic Peninsula region in the early 21st century, more significant in its northern part and the SSI [9]. In fact, our observation period of 2013–2017 overlaps with the last decade of meteorological records analyzed by such authors, for which they observed a summer average temperature change in the SSI of $-0.5$ °C with respect to the previous decade. This cooling is highly significant in this particular case, as melting modeling by Jonsell et al. [36] showed that such a temperature drop would imply a decrease in surface melting of 44%. The reason is that the hypsometry of the glacierized areas in the SSI is limited to a few hundred meters, and the summer average temperature is close to the melting point in large portions of the glaciers; therefore, a small temperature change implies a shift from non-melting to melting conditions or vice versa over large areas. The decrease in surface melting during the early 21st century has been confirmed at a wider scale (SSI and northern AP), e.g., by the study of Costi et al. [37]. The modeling by these authors, which spans the period 1981–2015, shows that surface melting in the SSI and the northern AP has decreased since the early 2000s. Both regions follow exactly the same pattern of temporal variations of specific surface melt, although it is more than two times larger in the SSI. However, while in the northern AP the specific surface melt and runoff are very similar in magnitude, indicating that nearly all melted snow and ice are lost by runoff, in the SSI, the magnitude of runoff is a large proportion (around two-thirds) of that of surface melt, indicating that a substantial amount of surface melt percolates and refreezes. As a result, the net specific summer surface losses are nearly equal in both regions. In parallel with the decreased summer melting during the early 21st century, a slight accumulation increase has been observed during such period [16] and attributed to a deepening of the circumpolar pressure trough, bringing a larger amount of moisture to the western part of the AP [38].

The available literature on geodetic mass balance estimates at particular locations in the SSI allows comparisons with our results. For instance, Osmanoglu et al. [13] estimated a total specific mass balance of $-0.67 \pm 0.40$ m w.e. a$^{-1}$ over the entire Livingston Island ice cap for the period 2007–2011. Our results for the same ice cap and the period 2013–2017 are $0.13 \pm 0.01$ m w.e. a$^{-1}$ (we took our result for an ice density of 900 kg m$^{-3}$, which was the one used by Osmanoglu et al.). Both results are compatible with the regional temperature evolution, as the period 2007–2011 had considerably warmer summers than the period 2014–2017 (summer averages of 0.4 vs. $-0.2$ °C, respectively, as recorded at Bellingshausen station, KGI [39], which has a good correlation—coefficient 0.82 [16]—with the summer temperature record at Juan Carlos I Station, Livingston Island). At a more local scale, the surface elevation changes during 2012–2016 of Ecology Glacier, KGI, observed by Pętlicki et al. [14], nearly overlap with our observation period of 2013–2017. Their average value for 2012–2016 was $-0.5 \pm 0.6$ m a$^{-1}$, which compares well (taking into account their relatively large error bounds) with our estimate of $-0.134 \pm 0.002$ m a$^{-1}$ for the period 2013–2017.

The analysis by Fieber et al. [15] includes, within the SSI, glaciers in both King George and Elephant Islands. The latter, however, is not covered by our study. Fieber's study

period 1956–2013 for the KGI glaciers immediately precedes our analyzed period 2013–2017 and, thus, allows us, with evident limitations due to the very distinct length of both time periods, to infer the evolution of the mass changes between them. As the average summer temperature for the period 2013–2017 measured at Bellingshausen station, KGI, was 0.5 °C, lower than that for the whole previous temperature record available, which started in 1968 (−0.2 vs. 0.3 °C [39]), we could expect a slightly less negative (or more positive) GMB between both periods. However, this is not necessarily the case, as the observation period by Fieber et al. [15] goes further back in the past, to 1956, when the temperatures were probably colder. Carefully verifying the results initially causes a level of deception. The reason is that Fieber's study considers nine glaciers in KGI, all of them covered by our study. However, the areal coverage (measured vs. total area) for three of such glaciers (Lange, Emerald Icefalls B, and Emerald Icefalls C) is lower than 60% (48, 59, and 44%, respectively) in Fieber's study, and in our study, for three other glaciers (Polar Committee Icefall, Urbanek Icefall, and Emerald Icefalls C), it is lower than 65% (64, 32, and 35%, respectively). However, it is not only those sets of glaciers with reduced coverage in each study that are non-overlapping, but, additionally, Fieber's reduced coverage focuses on the uppermost parts of the glaciers, while our reduced coverage is focused on their lowermost parts. The reason lies in the different measuring techniques employed. Whereas Fieber's study uses optical images, which often fail to properly resolve the accumulation zones because of their smooth snow surface, our study uses SAR images, which often fail in zones with steep slopes. Note that six of the nine KGI glaciers under consideration (the three remaining glaciers not cited so far are Znosko Glacier, Admiralen Glacier, and Emerald Icefalls A) have "Icefall" in their names, and the very steep slope icefalls are located at the lowermost part of the corresponding glaciers. The implication is that the poorly covered glaciers in each study are primarily sampled at different zones in each study, making any comparison of the results meaningless. Therefore, the comparison has to be restricted to Znosko Glacier, Admiralen Glacier, and Emerald Icefalls A, for which the coverage lies within 78–100% in both studies (in particular 100, 95, and 84%, respectively, for Fieber at al. and 90, 86, and 78%, respectively, in our study). For two of these glaciers, the GMB has indeed shifted to less negative or to slightly positive average values between the 1956–2013 and 2013–2017 periods: for the Znosko Glacier from −0.52 to 0.12 m w.e. a$^{-1}$ and for the Admiralen Glacier from −0.44 to −0.02 m w.e. a$^{-1}$ (we do not quote errors because they are very small, especially for Fieber's study, due to the long observation period involved; also note that we have recalculated the results of Fieber et al. using the same volume-to-mass conversion factor of 850 kg m$^{-3}$ employed in our study—they used 917 kg m$^{-3}$ in their original study). However, for Emerald Icefalls A, the shift was from slightly positive to negative GMB: from 0.03 m w.e. a$^{-1}$ for 1956–2013 to −0.48 m w.e. a$^{-1}$ for 2013–2017. The reason for the positive value for Emerald Icefalls A during 1956–2013 was not a glacier advance, as the glacier in fact retreated locally by a maximum of 70 m [15]. A similar situation, but with an opposite sign, was observed by Fieber et al. [15] for the Znosko Glacier, which experienced considerable surface lowering but advanced locally by almost 100 m over the 56-year period. For the rest of their analyzed glaciers, volume loss/gain was correlated with a respective retreat/advance. Two lessons are to be learnt from this comparison of results and are worth noting even if known by many: (1) volume loss/gain is not necessarily correlated with a respective retreat/advance, and (2) it is convenient, whenever possible, to combine optical-based and radar-based images to obtain a more complete coverage of the glaciers under study.

Our geodetic mass balance estimates are also fairly consistent with those obtained by the glaciological method in the limited set of locations where the latter are available. During our analyzed period of 2013–2017, SMB observations by the glaciological method at a glacier basin scale are only available for Hurd and Johnsons Glaciers in Livingston Island [16]. As Hurd is a land-terminating glacier, its SMB should be comparable to the GMB (assuming negligible internal and basal mass balances). Johnsons, however, is a tidewater glacier; therefore, the calving losses have to be added to the SMB before comparison with the

GMB. Hurd and Johnsons average total mass balances for the hydrological years 2015–2017 (remember that the hydrological year of the southern hemisphere is the one-year period ending on 31st of March) were of 0.15 and 0.27 m w.e. a$^{-1}$, respectively (the latter after subtracting 0.14 ± 0.04 m w.e. a$^{-1}$ for the calving losses [16]). The year 2015 had the most positive SMB of this MB series (started in 2002), for which the ELA was at sea level and the accumulation area ratio (AAR) was of 100% for both glaciers. The year 2016 was the last year in a series of seven consecutive years with positive SMBs and 2017 the first with a negative SMB following such a positive series. The average ELA and AAR for 2015–2017 were 125 m and 73% (Hurd) and 98 m and 85% (Johnsons). The average total mass balances of 0.15 and 0.27 m w.e. a$^{-1}$ for Hurd and Johnsons compare well with the GMB of 0.11 ± 0.01 m w.e. a$^{-1}$ obtained in this study for Livingston Island (Table 1) taking into account that a typical error bar for the SMB calculated by the glaciological method is of 0.1 m w.e. a$^{-1}$, in addition to the uncertainty involved in the calving estimate.

The mass balance changes are not homogeneously distributed but are strongly influenced by the hypsometry. In general, the largest losses are concentrated at low elevations (Figure 3i), where the error bars are largest, and decrease with increasing elevations, occasionally stabilizing at a certain altitude. Of the largest islands, Livingston (Figure 3a) has its larger mass gains in elevation bins with a large share of the total area, thus resulting in positive total mass balance. Although the higher elevation bins show the largest error bars of $\Delta h / \Delta t$, these values do not contribute significantly to the total error due to the small amount of glacierized area at high elevations. King George Island, which has a more homogeneous hypsometry, shows a steady decrease in mass losses, stabilizing at values close to balance at around 350 m altitude. Considering the full set of islands under study, the mass losses at low elevations combined with stable or mass gains at high elevations contribute to a mass imbalance, which is more marked for the land-terminating glaciers due to their low velocities.

## 5. Conclusions

Our study presents a contribution to fill an important observational gap in the vicinity of research on the northern Antarctic Peninsula. The following main conclusions can be drawn from our analysis:

(1) For the period 2013–2017, the South Shetland Islands showed a mean surface elevation change rate of −0.125 ± 0.004 m a$^{-1}$, equivalent to a specific mass balance of −0.106 ± 0.007 m w.e. a$^{-1}$ and representing a total mass change rate of −0.238 ± 12 Gt a$^{-1}$.

(2) Our specific mass balance result, close to balance, is in agreement, within their error bounds, with the results of Hugonnet et al. [2] for the Antarctic and Subantarctic regions during the period 2015–2019 and also consistent with those of Zemp et al. [21] for the longer period of 2006–2016.

(3) Our specific mass balance results are also in agreement with glacier basin-scale observations by the geodetic and the glaciological methods at particular locations in the South Shetland Islands during our study period.

(4) The observed changes are also compatible with the hiatus of warming and the temporary cooling trend observed in the northern part of the Antarctic Peninsula and the South Shetland Islands during the first 15 years of the present century. Taking into account the regional temperature evolution, our specific mass balance results are consistent with those of other glacier basin-scale studies in the region for non-overlapping or partly overlapping periods.

Although the current estimated regional mass losses are modest, they have been projected to strongly increase until the end of the 21st century [3], when the Antarctic and Subantarctic regions are expected to be among the largest contributors to sea level rise. This emphasizes the interest of monitoring the evolution of mass balance of the peripheral Antarctic glaciers on a regular basis, including those of the South Shetland Islands, and highlights the need for, e.g., a more recent coverage by TanDEM-X data. Since the recent

regional cooling seems to have come to an end [10], we are at a particularly critical moment to reinforce the observational effort in this region.

**Author Contributions:** Conceptualization, K.S., T.S., and F.N.; methodology, K.S., T.S., and C.S.; software, K.S. and T.S.; formal analysis, K.S. and F.N.; data curation, K.S.; writing—original draft preparation, K.S. and F.N.; writing—review and editing, T.S. and M.B.; visualization, K.S.; supervision, F.N., T.S., and M.B.; funding acquisition, F.N. All authors have read and agreed to the published version of the manuscript.

**Funding:** This research was funded by Agencia Estatal de Investigación, grant numbers CTM2017-84441-R and PID2020-113051RB-C31.

**Data Availability Statement:** Our basic data come from TanDEM-X scenes listed in Table S1. The main results are presented in the Table and Figures of the paper. Elevation change fields are available via the World Data Center PANGAEA operated by AWI Bremerhaven (after publication, DOI is requested).

**Conflicts of Interest:** The authors declare no conflict of interest.

## Appendix A

**Table A1.** TanDEM-X/TerraSAR-X data information used for the GMB in SSI.

| Island | 2017 Data Used | | | 2014 Data Used (2013 for KGI) | | |
|---|---|---|---|---|---|---|
| | Date | Path Number | Strip Number | Date | Path Number | Strip Number |
| Livingston | 19 May 2017<br>30 May 2017 | 110<br>110 | 0005<br>0015 | 4 August 2014<br>15 August 2014 | 013<br>013 | 0040<br>0050 |
| Deception | 19 May 2017 | 110 | 0005 | 23 June 2014 | 034 | 0045 |
| Robert | 27 June 2017 | 034 | 0035 | 2 July 2014 | 013 | 0060 |
| Greenwich | 27 June 2017 | 034 | 0035 | 2 July 2014 | 013 | 0060 |
| Nelson | 16 June 2017 | 034 | 0025 | 21 May 2014 | 034 | 0025 |
| King George | 25 May 2017<br>16 June 2017 | 034<br>034 | 0015<br>0025 | 1 July 2013<br>9 June 2013<br>20 June 2013 | 125<br>125<br>125 | 0040<br>0050<br>0060 |
| Snow | 21 June 2017 | 110 | 0025 | 4 July 2014 | 034 | 0055 |
| Low | 21 June 2017 | 110 | 0025 | 4 May 2014 | 110 | 0025 |

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
