# Peer review of "Geodetic Mass Balance of the South Shetland Islands Ice Caps, Antarctica, from Differencing TanDEM-X DEMs"

_remotesensing, doi:10.3390/rs13173408_

Round 1
Reviewer 1 Report
The paper is well written and informative. The level of novelty in the used method is moderate. However, the paper could be interesting for a reader to have a clearer understanding of the benefits in using satellite-based technologies to detect the mass balance change and make predictions on the future evolution of SLR. Of course, the paper is informative but it does not provide a clear path on potential applications of these important finding.
Author Response
Dear reviewer,
Please find attached our rebuttal letter of the first round revision of our paper.
Sincerely,
Kaian Shahateet

Reviewer 2 Report
I find this work an important addition to the literature on the South Shetland ice dome. The paper is well written and well structured. However, I have my doubts about the generally poor discussion and introduction, as well as the inaccuracies found in this work. It seems as if it is a result of some haste and incomprehensible omissions of important information concerning the region as well as the topic itself, which are easy to find in the available literature.
Already in the abstract, as well as in the paper itself, the authors write that they omit Ivory Coast from their analysis. This is completely incomprehensible in my view, since the ice dome of this island, was the subject of a paper published in 2018 that was relevant to the topic, and it seems appropriate to at least refer to its results in the discussion or introduction itself, even if the authors do not intend to subject it to analysis.
See:
Fieber et al. (2018). Remote Sensing of Environment, 205, 18-31.
Nor is the message of the first paragraph of the introduction clear, which sets the reader quite subjectively to the trends occurring in the South Shetland area. Meteorological data, from 2013-2017, available from the Polish Antarctic Station, which, unlike most of the other meteorological stations, is located inside King George Island, contradict the reversal of the trend in mid-2010 and indicate a longer persistence of the cooling effect during the summer.
See:
Plenzler et al. (2019). Polish Polar Research, 40, 1-27.
I also cannot agree with the sentence in line 72, i.e., "whole island ice cap (KGI)", as I am not aware of any glaciological work on the eastern part of the Arctowski Dome or the Krakow Ice Dome. This should be verified.
I also do not understand the lack of reference to the research carried out on the Ecology glacier on King George Island or Greenwich Island (the only ones of their kind) by Dr. Pętlicki's team. There is no reference to them also in paper 14, indicated as the one in which similar research was summarized. See lines 64-65.
See:
Pętlicki and Kinnard (2016). Journal of Glaciology, 62.
Pętlicki et al. (2017). Remote Sensing, 9, 520.
So, in my opinion, the work needs to be rewritten as far as the introduction is concerned, and quite substantially. Also, the discussion needs significant improvement.
Author Response
Dear reviewer,
Thank you very much for your contribution. Please find attached our rebuttal letter of the first round revision of our paper.
Sincerely,
Kaian Shahateet

Reviewer 3 Report
Dear Authors & Editors,
Thank you very much for the interesting and well-presented manuscript. I have very little to suggest in order to improve the manuscript, but there are a few things i would like to mentions.
In figure 2 it is relatively hard to se surface change patterns. This could most likely be improved by choosing another color-scale and showing a larger image. In addition, the legend is unclear, i guess the colors are showing ranges of several meters? This should be clarified.
In figure 4 some writing is very hard to read. This might be a formatting issue but it would be great to have this figure more readable, either by enlarging it or by changing the written parts.
Best regards,
Author Response

(The authors gave the same response as above.)

Round 2
Reviewer 2 Report
Thank you very much for your comprehensive reply. I would like to apologise for the misunderstanding in my review. I made a mistake regarding the name in Elephant Island, which is sometimes colloquially referred to in my country as Ivory Coast. This is my mistake for which I apologise. However, the authors, unaware of the slip of the tongue I committed, have corrected the paper very well taking into account the article by Fieber et al.
I believe that all corrections have been taken into account and the paper can now be published without undue delay. Congratulations on your research.